# Learning to Generate Inversion-Resistant Model Explanations

**Hoyong Jeong, Suyoung Lee, Sung Ju Hwang, Sooel Son**
KAIST
{yongari38, suyoung.lee, sjhwang82, sl.son}@kaist.ac.kr

## Abstract

The wide adoption of deep neural networks (DNNs) in mission-critical applications has spurred the need for interpretable models that provide explanations of the model's decisions. Unfortunately, previous studies have demonstrated that model explanations facilitate information leakage, rendering DNN models vulnerable to model inversion attacks. These attacks enable the adversary to reconstruct original images based on model explanations, thus leaking privacy-sensitive features. To this end, we present Generative Noise Injector for Model Explanations (GNIME), a novel defense framework that perturbs model explanations to minimize the risk of model inversion attacks while preserving the interpretabilities of the generated explanations. Specifically, we formulate the defense training as a two-player minimax game between the inversion attack network on the one hand, which aims to invert model explanations, and the noise generator network on the other, which aims to inject perturbations to tamper with model inversion attacks. We demonstrate that GNIME significantly decreases the information leakage in model explanations, decreasing transferable classification accuracy in facial recognition models by up to 84.8% while preserving the original functionality of model explanations.

## 1 Introduction

The recent widespread adoption of deep neural networks (DNNs) in building practical real-world applications has led to a surge of interest in explainable AI (XAI) since mission-critical tasks (e.g., medical diagnosis) often request explanations of the model's predictions. Such explanations are not only helpful in debugging the model but also in providing explanations to end-users on how predictions are made. They are often provided in the form of a feature attribution map, which shows the importance of each feature computed based on the gradients of a target model.

Unfortunately, recent studies have demonstrated that such explanations cause additional information leakage, which can be exploited by an adversary [1, 8, 19, 22, 32]. Specifically, Zhao *et al.* [32] presented a novel black-box model inversion (MI) attack which reconstructs an input image given its prediction vector and the model explanation from a target DNN model. They demonstrated that additional information leaked through model explanations (e.g., Grad or Grad-CAM) increases the performance of MI attacks by a factor of 2.4 in terms of transferable classification accuracy, enabling high-fidelity attacks.

Figure 1 compares the result of MI attacks using model predictions alone (PredMI) and explanation-aware MI attacks using model predictions and explanations together (ExpMI). The examples reconstructed via ExpMI show high-quality faces compared to those via PredMI, leaking additional information, such as facial expressions and accessories. However, previous works have only proposed the attacks exploiting the information leakage, without providing any defenses against them.

36th Conference on Neural Information Processing Systems (NeurIPS 2022).

Thus, to defend against MI attacks exploiting model explanations, we propose GNIME, which generates inversion-resistant model explanations. Figure 2 shows the overall GNIME pipeline. GNIME consists of two phases: training and deployment. In the training phase, GNIME trains a noise generator that injects perturbations into the explanations provided by the model under protection to minimize the MI threat. To train this noise generator, GNIME trains an MI attack network to compete against each other. Then, in the deployment phase, we deploy the trained noise generator to obfuscate the explanations generated for each test instance.

The key idea of our approach is to establish a two-player minimax game between the MI attack (INV) network and the noise generator (NG) network. We design the NG network to inject perturbations into the model's explanations (i.e., attribution map); the noise-injected model explanation is then fed into the INV network to reconstruct the inverted image. GNIME conducts an alternating training of these networks. During the NG's turn, the NG network is trained to tamper with the inversion performance of the INV network while minimizing the perturbations to the model explanations. During the INV's turn, we train the INV network to maximize its model inversion performance. These alternating training procedures improve the INV network to better invert the given model's explanations while the NG network learns to inject perceptually imperceptible inversion-resistant perturbations to the provided explanations. After training is complete, the GNIME owner deploys the NG network to inject perturbations into the original explanations before releasing them.

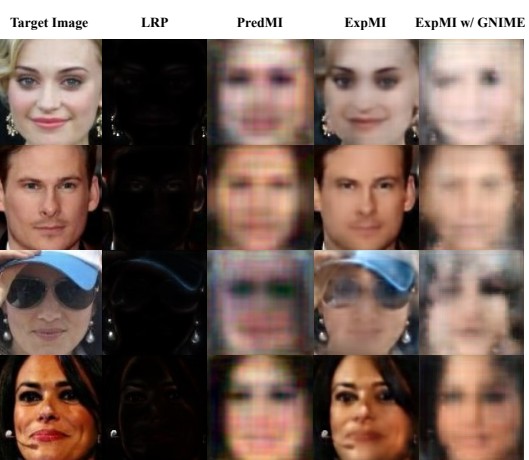

Figure 1: Qualitative comparison of the face reconstruction results of PredMI and ExpMI attacks using LRP explanations. The last column shows the ExpMI attack results when a target ML system is under protection with GNIME.

We evaluate the efficacy of GNIME, by measuring 1) how effective the model is in decreasing the risk of information leakage from the model explanations and 2) how well the perturbed explanations preserve the semantics of the original explanations. For different types of explanations (i.e., Grad [24], Grad-CAM [21], and LRP [3]), we demonstrate that GNIME consistently generates infinitesimal inversion-resistant noises that significantly impede explanation-aware MI attacks. To facilitate further research, we publish GNIME at `https://github.com/WSP-LAB/GNIME`.

In summary, our contributions are as follows:

- We propose the very first defense framework against explanation-aware MI attacks, which learns to suppress inversion-critical features from model explanations.
- We evaluate GNIME on diverse datasets and demonstrate its efficacy in decreasing the MI risk while preserving the interpretability of the original explanations.

## 2   Related Work

**Explainable AI for CNN models.** Previous XAI research has introduced diverse methods of computing an attribution map (i.e., saliency map) that explains the decision of a DNN classifier. Specifically, given an input image, its attribution map highlights important features that heavily influence the classifier's decision. A straightforward approach is to calculate the gradient of the DNN model's output with respect to each input pixel [24], which we refer to as Grad. Grad thus produces an attribution map of which the size is identical to that of input images. The larger the absolute Grad value of a pixel, the greater influence the corresponding pixel has in deriving a model's decision. Numerous variants have been proposed to supplement or extend the explainability of Grad [23, 25, 26]. In this paper, we use Grad as a basic type of for model explanations. The class activation map (CAM) [33] is also a prevalent means of computing model explanations. It provides a coarse-grained activation map that represents the state of intermediate convolutional layers. Gradient-weighted CAM or Grad-CAM [21] is an extension of CAM. Unlike CAM, Grad-CAM does not require a specific model architecture. It performs partial back-propagation until reaching the final

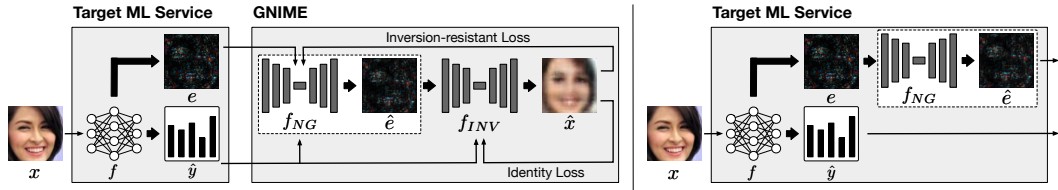

Figure 2: Overview of the GNIME pipeline.

convolutional layer before the fully-connected (FC) layers. The activation maps are then aggregated by globally pooling all existing channels, thus constituting a heatmap, which can be overlaid upon the original image. The layer-wise relevance propagation (LRP) takes a holistic approach to explaining a model's decision [3]. It performs importance decomposition by redistributing relevance scores along the backward pass and derives the relative contribution of each pixel of the input image.

**Model Inversion Attacks.** An MI attack refers to an attempt to invert a given model's output, thus computing its approximate input. The first MI attack targeted simple linear models to reconstruct certain genetic markers from medical dosage predictions [10]. Fredrikson *et al.* [9] expanded this idea to target other types of machine learning (ML) algorithms, such as decision trees and shallow neural networks, which enable the attack to target facial recognition models. These studies leveraged the maximum a posteriori principle to construct input features that maximize the likelihood of a given model's response. However, their reconstructed images were blurry and of low-resolution; these methods are unfit for reconstructing input images against complicated DNN models.

Yang *et al.* [29] proposed the idea of utilizing a dedicated DNN model to perform MI attacks. They pretrained this inversion model with inputs that the attacker generates and their corresponding outputs from a target DNN model. This training-based MI attack drastically improved the model inversion performance. Follow-up works have demonstrated the use of generative adversarial networks (GANs) to improve the inversion performance [2, 5, 27, 31]. Recently, Zhao *et al.* [32] showed that the additional information leakage from XAI models can be further exploited to enhance the effectiveness of the MI attack. While previous GAN-based attacks focus on producing generic images of a target class, the explanation-aware attack by Zhao *et al.* [32] is able to reconstruct features specific to test instances, such as face images with hats, glasses, or earrings. However, to our knowledge, no defense mechanism has been proposed to defend against such explanation-aware MI attacks.

## 3 Explanation-aware Model Inversion Attack

Previous studies have demonstrated the critical threat that generative MI attacks impose. In this section, we describe the threat model for conducting state-of-the-art generative MI attacks against which GNIME is designed to protect model explanations. We then explain the state-of-the-art explanation-aware MI attacks using generative networks along with notations.

### 3.1 Threat Model

We consider a target classifier $f$ providing black-box access whereby an adversary can query $f$ to obtain its output $\hat{y}$ in the form of a prediction vector. Each vector value indicates the predicted probability for the corresponding class. Furthermore, the target service leveraging $f$ also provides a model explanation $e$ (e.g., a Grad, Grad-CAM, or LRP saliency map) that illustrates important features contributing to $f$ emitting $\hat{y}$.

We assume a man-in-the-middle adversary who observes a pair of $e$ and $\hat{y}$ for an input image inaccessible to the adversary. Thus, the adversary's goal is to reconstruct a high-fidelity input image given $e$ and $\hat{y}$. Furthermore, we assume that the adversary is able to obtain an auxiliary dataset $D_{aux}$ whose underlying data distribution is similar to that of the training data $D$ for the target classifier $f$. The adversary leverages this auxiliary dataset to train her inversion networks $f_A$. Since the adversary can collect $(\hat{y}, e)$ by querying $f$ for each input $x \in D_{aux}$, she exploits both $\hat{y}$ and $e$ to train $f_A$.

Note that our adversary model is analogous to that of Zhao *et al.* [32], which constitutes a strong black-box adversary model. Also, Yang *et al.* [29] assumes a similar man-in-the-middle adversary

who reconstructs input images from prediction vectors observed from $f$. Our goal is to protect the model explanations from this strong black-box adversary conducting MI attacks.

## 3.2 Model Inversion Attacks Using Model Explanations

Zhao *et al.* [32] proposed a novel black-box MI attack (ExpMI) that exploits prediction vectors generated from a target classifier $f$ as well as the model explanations for each decision. To reconstruct the input images, they trained a transposed CNN model $f_A$ based on auxiliary data, each of which was a prediction vector paired with its model explanation.

Consider a target classifier $f$ that is a function $f(x) \to (e, \hat{y})$ that emits a prediction vector and model explanation (i.e., an attribution map). The adversary's goal is to compute $f_A(e, \hat{y}) \to \hat{x}$, which takes explanation $e$ and $\hat{y}$ to produce $\hat{x}$, revealing sensitive attributes of $x$. Figure 3 illustrates a simplified structure of the ExpMI model. To take advantage of structural information in activation maps, ExpMI utilizes a U-Net-style bypass connection along with flattened inputs directly appended to the intermediate embedding.

The adversary takes advantage of $e$, which leaks sensitive information pertaining to $x$ and drastically improves the reconstruction performance of $f_A$. As Figure 1 shows, the quality of the reconstructed image $\hat{x}$ with $e$ is significantly better than the MI attack using prediction vectors alone generates.

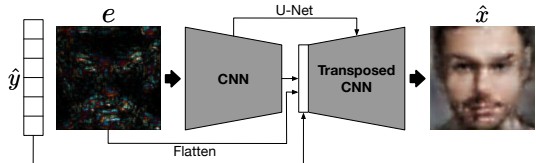

Figure 3: DNN architecture of ExpMI models.

Considering that model explanation has become instrumental in identifying the reasons behind the decisions of DNN models in various industry sectors, it becomes inevitable for corporations to provide model explanations. However, providing such information also jeopardizes privacy by revealing sensitive attributes of reconstructed images. To this end, we propose GNIME, a novel defense framework that learns to inject noise into model explanations, thus minimizing the privacy risk of ExpMI attacks and maximizing the utility of noise-injected model explanations.

## 4 Learning to Generate Noise for Model Explanations

Figure 2 depicts the overall workflow of GNIME. Phase I trains the noise injector and MI attack networks, thus inducing the noise injector network to inject inversion-resistant perturbations into model explanations. Phase II then deploys the noise injector network in injecting noise into each model explanation upon its release.

Given a target classifier $f(x) \to (e, \hat{y})$, GNIME learns to inject noise into $e$ to minimize the MI threat. Our approach utilizes the noise generator network ($f_{NG}$) and inversion network ($f_{INV}$), both of which are trained upon $D_T = \{x_i, y_i\}_{i=1}^N$, the original training data of $f$.

Let $f_{NG}(e, \hat{y}) \to \hat{e}$ be the NG network that generates a new explanation attribution map $\hat{e}$, which includes noise. For the INV network, we design $f_{INV}(e, \hat{y}) \to \hat{x}$ that takes $e$ and $\hat{y}$, then computes an inversion image $\hat{x}$. For the loss functions of $f_{NG}$ and $f_{INV}$, we use an explanation distortion loss ($\mathcal{L}_{xd}$) to encourage minimum perturbation to the explanations and an image reconstruction loss ($\mathcal{L}_{re}$) to encourage inversion performance. We use the pixel-wise mean squared error (MSE) between $e$ and $\hat{e}$ for $\mathcal{L}_{xd}$, and the MSE between $x$ and $\hat{x}$ for $\mathcal{L}_{re}$.

Inspired by Generative Adversarial Networks [11], we form a two-player minimax game in which $f_{NG}$ and $f_{INV}$ are trained together using two different objectives. Specifically, the training phase consists of two steps for each epoch.

In the first step, we freeze $f_{NG}$ and train $f_{INV}$ by minimizing the following loss:

$$\mathcal{L}_{INV} = \mathcal{L}_{re}(x, f_{INV}(e, \hat{y})) + \mathcal{L}_{re}(x, f_{INV}(\hat{e}, \hat{y})), \; where \; \hat{e} = f_{NG}(e, \hat{y}). \tag{1}$$

The objective is to improve the inversion capability of $f_{INV}$ given $(e, \hat{y})$ and $(\hat{e}, \hat{y})$. Note that $\mathcal{L}_{INV}$ includes the reconstruction loss from the clean explanations in order to prevent the network from overfitting to the perturbed explanations. For the second step, we freeze $f_{INV}$ and train $f_{NG}$ by minimizing the following loss:

$$\mathcal{L}_{NG} = \mathcal{L}_{xd}(e, \hat{e}) - \lambda \mathcal{L}_{re}(x, f_{INV}(\hat{e}, \hat{y})). \tag{2}$$

---
**Algorithm 1** Training algorithm in Phase I
---
**Input:** target classifier $f$ and input data $X = \{x_1, ..., x_n\}$
**Output:** noise generator network $f_{NG}$
Generate $\hat{Y}$ and $E$ from each input $X$ to $f$
**for** number of epochs **do**
    Sample a batch of size $m$: $\{(x_1, \hat{y}_1, e_1), ..., (x_m, \hat{y}_m, e_m)\}$, consisting of pairs from $X$, $\hat{Y}$, and $E$
    Let $\hat{e}_i = f_{NG}(e_i, \hat{y}_i)$ in

        Update $f_{INV}$:    $\nabla_{\theta_{f_{INV}}} \frac{1}{m} \sum_{i=1}^{m} [\mathcal{L}_{re}(x_i, f_{INV}(e_i, \hat{y}_i)) + \mathcal{L}_{re}(x_i, f_{INV}(\hat{e}_i, \hat{y}_i))]$
        Update $f_{NG}$:    $\nabla_{\theta_{f_{NG}}} \frac{1}{m} \sum_{i=1}^{m} [\mathcal{L}_{xd}(e_i, \hat{e}_i) - \lambda \mathcal{L}_{re}(x_i, f_{INV}(\hat{e}_i, \hat{y}_i))]$
---

It aims to minimize the distortion between $e$ and $\hat{e}$, while maximizing the error between $x$ and $\hat{x}$. Therefore, we train $f_{NG}$ to generate a noise-injected attribution map that is similar to the original attribution map but causes $f_{INV}$ to generate an inversion image dissimilar from its original image. Algorithm 1 explains our training process. In $\mathcal{L}_{NG}$, we set $\lambda = 500$ for CelebA models and $\lambda = 100$ for MNIST, CIFAR-10, and ImageNet-100 models, then deploy the final model after 500 epochs.

In the worst case, $f_{NG}$ will merely cause a regularization effect on $f_{INV}$, rendering it robust to any noisy explanation. However, as $f_{NG}$ is trained over epochs, it learns to capture and suppress features inside explanations that contain exploitable information regarding the original input image. Once the training is finished, we discard $f_{INV}$ and only make use of $f_{NG}$ for Phase II. For every attribution map $e$, we use $f_{NG}$ to generate a noise-injected attribution map $\hat{e}$, then release $\hat{e}$ instead of $e$ upon a given query $x$.

Additionally, GNIME service providers can inject less noise into the explanations with an arbitrary MSE bound, $\gamma$. In this case, GNIME computes a new attribution map $e'$ (see Equation 3); we normalize the noise difference $n$ between $\hat{e}$ and $e$ by dividing it by the square root of the variance of $n$ over $\gamma$ and then add this normalized difference to $e$:

$$e' = e + \frac{n}{\sqrt{\sigma^2(n)/\gamma}}, \ where \ n = \hat{e} - e. \tag{3}$$

$\sigma^2(n)$ denotes the pixel variance of $n$. Therefore, by controlling $\gamma$, GNIME service providers can utilize the trade-off and choose to preserve better explanation functionality in exchange for defensive capability. Refer to the supplementary material for further analysis regarding the trade-off.

## 5 Experiments

### 5.1 Experimental Setup

**Datasets.** We evaluate GNIME using four datasets: (1) CelebA [17], (2) MNIST [14], (3) CIFAR-10 [13], and (4) ImageNet [6] , each of which is freely available for research purposes. For CelebA, we use its balanced subset consisting of 30,000 face images of 1000 individuals. To obtain a tighter bound for the faces, we crop the center and resize each CelebA image to $128 \times 128$. MNIST consists of 70,000 images of handwritten digits, and CIFAR-10 has 60,000 images of 10 different objects. Both MNIST and CIFAR-10 images were resized to $32 \times 32$. We note that previous studies [29, 31, 32] adopted the same datasets to demonstrate how effectively their frameworks mitigate the MI threat. In addition, we evaluate GNIME on ImageNet-100 [6], which is a large scale dataset of higher resolution, consisting of 100 classes randomly selected classes from ImageNet; each image is resized to $256 \times 256$.

We split each dataset into two disjoint sets of the same size: the target and attack datasets. The target dataset is used to train a target model $f$ and GNIME. The attack dataset is used to train the inversion networks for explanation-aware MI attacks and to test GNIME. Specifically, we split this attack dataset with an 80/20 ratio for the train/test split.

Note that we assume an adversary capable of training $f_A$ upon a dataset of which the underlying distribution is nearly identical to that of the target network's training data. We intentionally assume this strong adversary to demonstrate how good GNIME is in mitigating the MI threat. Moreover, we also assess the performance of GNIME against a weaker but more practical adversary using an auxiliary dataset that has a different underlying distribution (i.e., a separately collected set for

conducting a similar task) [29]. For instance, when attacking the CelebA model, we randomly sample face images from FaceScrub [20] to train $f_A$. Note that the overlapping identities from FaceScrub which also appear in CelebA are removed before the sampling process. We extract a tighter bound of each FaceScrub face according to its official bounding box information.

**Target Models.** We implement different target models for different datasets. For CelebA, the target model consists of three convolutional layers followed by two fully-connected (FC) layers. We employ dropout before each FC layer to avoid overfitting. For MNIST and CIFAR-10, we use a target model with two convolutional layers followed by two FC layers. We selected the same models that Zhao *et al.* [32] used to demonstrate their MI attacks. For ImageNet-100, we used the model from CelebA with an additional convolutional layer.

**Inversion Methods.** We use two MI attack methods (ExpMI and PredMI) as baselines in our experiments. Specifically, we compare the efficacy of various defense mechanisms against the state-of-the-art ExpMI attack [32]. In addition, we consider the prediction-only MI attack using no model explanations (PredMI), suggested by Yang *et al.* [29]. Note that the inversion results via PredMI form an upper bound that represents the ideal case for any defenses in which model explanations leak no additional knowledge that the adversary is able to exploit. We conducted each MI attack five times and reported average metrics with standard deviations. All experiments took place on a system equipped with 512GBs of RAM, two Intel Xeon Gold 6258R CPUs, and four RTX 3090 GPUs.

## 5.2 Evaluation Metrics

We assess whether the reconstructed image is close to its original image by leveraging four metrics: MSE, SSIM, TCA, and DeePSiM, which have been extensively used for quantitative evaluations in previous studies [29, 31, 32].

**Mean Squared Error (MSE).** The MSE is a metric that measures the distance between a reconstructed image and its corresponding original image. A small MSE indicates that the reconstructed image is close to its original image. For normalization, input pixel values of the inputs are scaled to be within the range of $[0, 1]$.

**Structural Similarity Index Measure (SSIM).** The SSIM computes the distortion of structural information between two images [28]. Since humans are good at recognizing structural differences between two images, this metric shows human perceptual similarity between two images. An SSIM score spans from 0 to 1, and the higher score indicates better reconstruction.

**Deep Perceptual Similarity Metric (DeePSiM).** The DeePSiM shows the similarity between two images at an intermediate representation level [7]. Unlike SSIM, which illustrates perceptual similarity, DeePSiM explains perceptual similarity from the neural network perspective. Specifically, we trained a separate evaluation classifier $f_E$, which shares the same architecture and training dataset with the target model. We then computed the feature distance between intermediate results from the penultimate layer of $f_E$, comparing the high-level features identified by $f_E$ [32]. If $f_{INV}$ accurately reconstructs the high-level image features, it results in a high DeePSiM score.

**Transferable Classification Accuracy (TCA).** We also measure a TCA to evaluate whether inverted images are generally recognizable by another DNN classifier of which the structure is the same as that of the target model. This metric indirectly shows whether the reconstructed image incorporates the key features of its original image. We measured this metric using $f_E$. Note that $f_E$ would correctly predict the reconstructed image if the reconstructed image successfully incorporates the key features of the original image, rather than to overfit to unimportant details captured only by the target classifier.

## 5.3 Experiment Results

### 5.3.1 Defending against MI Attacks

We evaluate the performance of vanilla explanation-aware MI attacks and then measure the efficacy of GNIME in decreasing information leakage from such MI attacks. Specifically, we compare five settings (two baseline attacks and three defenses against ExpMI attacks): (1) MI attacks using the model's prediction outputs alone (PredMI); (2) MI attacks using both prediction outputs and model explanations (ExpMI); (3) ExpMI attacks against a defense that injects Gaussian noise into saliency maps (RND); (4) ExpMI attacks against a defense that injects optimized noise into saliency maps

Table 1: Quantitative comparison between five different evaluation settings: PredMI, ExpMI, RND, OND, and GNIME. Values inside brackets indicate the degradation in inversion performance compared to ExpMI. The best results in mitigating ExpMI are highlighted in bold.

| | Metric | PredMI | Grad | | | | Grad-CAM | | | | LRP | | | |
|---|---|---|---|---|---|---|---|---|---|---|---|---|---|---|
| | | | ExpMI | RND | OND | GNIME | ExpMI | RND | OND | GNIME | ExpMI | RND | OND | GNIME |
| **CelebA** | MSE↑ | ±.0015
.0287 | ±.0011
.0141 | ±.0010
.0184
(.0043↑) | ±.0009
.0185
(.0045↑) | ±.0005
**.0222**
(**.0081↑**) | ±.0011
.0183 | ±.0006
.0191
(.0009↑) | ±.0003
.0200
(.0017↑) | ±.0004
**.0229**
(**.0046↑**) | ±.0008
.0069 | ±.0002
.0124
(.0056↑) | ±.0011
.0155
(.0086↑) | ±.0007
**.0234**
(**.0165↑**) |
| | SSIM↓ | ±.0242
.4427 | ±.0276
.6504 | ±.0302
.5430
(.1074↓) | ±.0216
.5357
(.1147↓) | ±.0060
**.5093**
(**.1410↓**) | ±.0189
.5823 | ±.0144
.5417
(.0406↓) | ±.0080
.5274
(.0550↓) | ±.0046
**.4952**
(**.0871↓**) | ±.0224
.8233 | ±.0022
.6370
(.1863↓) | ±.0209
.5805
(.2428↓) | ±.0130
**.4960**
(**.3273↓**) |
| | TCA↓ | ±.0050
.0217 | ±.0142
.1000 | ±.0171
.0690
(.0310↓) | ±.0114
.0570
(.0430↓) | ±.0032
**.0254**
(**.0746↓**) | ±.0085
.1097 | ±.0067
.0860
(.0237↓) | ±.0034
.0795
(.0302↓) | ±.0034
**.0445**
(**.0652↓**) | ±.0062
.2147 | ±.0037
.1559
(.0588↓) | ±.0116
.1207
(.0940↓) | ±.0103
**.0326**
(**.1821↓**) |
| | DeepPSiM↓ | ±.0033
.1851 | ±.0155
.2470 | ±.0111
.2117
(.0353↓) | ±.0095
.2064
(.0406↓) | ±.0021
**.1880**
(**.0590↓**) | ±.0136
.2745 | ±.0063
.2556
(.0189↓) | ±.0031
.2490
(.0255↓) | ±.0027
**.2040**
(**.0706↓**) | ±.0284
.4922 | ±.0034
.3678
(.1243↓) | ±.0161
.3006
(.1915↓) | ±.0092
**.1944**
(**.2978↓**) |
| **MNIST** | MSE↑ | ±.0001
.0237 | ±.0003
.0051 | ±.0002
.0067
(.0016↑) | ±.0003
.0070
(.0019↑) | ±.0003
**.0143**
(**.0092↑**) | ±.0003
.0021 | ±.0002
.0059
(.0038↑) | ±.0003
.0061
(.0040↑) | ±.0002
**.0177**
(**.0156↑**) | ±.0001
.0012 | ±.0001
.0037
(.0025↑) | ±.0003
.0044
(.0032↑) | ±.0001
**.0168**
(**.0156↑**) |
| | SSIM↓ | ±.0068
.5542 | ±.0133
.9065 | ±.0249
.8707
(.0358↓) | ±.0115
.8753
(.0312↓) | ±.0046
**.7686**
(**.1379↓**) | ±.0068
.9613 | ±.0039
.9009
(.0605↓) | ±.0150
.8929
(.0685↓) | ±.0026
**.7213**
(**.2400↓**) | ±.0009
.9767 | ±.0052
.9314
(.0453↓) | ±.0064
.9044
(.0723↓) | ±.0037
**.7356**
(**.2411↓**) |
| | TCA↓ | ±.0024
.9751 | ±.0009
.9850 | ±.0005
.9835
(.0015↓) | ±.0012
.9826
(.0024↓) | ±.0011
**.9770**
(**.0080↓**) | ±.0009
.9871 | ±.0010
.9825
(.0047↓) | ±.0010
.9815
(.0056↓) | ±.0021
**.9715**
(**.0156↓**) | ±.0004
.9867 | ±.0012
.9833
(.0033↓) | ±.0008
.9815
(.0052↓) | ±.0014
**.9726**
(**.0141↓**) |
| | DeepPSiM↓ | ±.0009
.8541 | ±.0025
.9367 | ±.0010
.9317
(.0049↓) | ±.0020
.9293
(.0074↓) | ±.0022
**.8974**
(**.0392↓**) | ±.0027
.9645 | ±.0024
.9358
(.0287↓) | ±.0024
.9353
(.0292↓) | ±.0019
**.8850**
(**.0794↓**) | ±.0008
.9695 | ±.0007
.9505
(.0190↓) | ±.0056
.9287
(.0409↓) | ±.0017
**.8874**
(**.0821↓**) |
| **CIFAR-10** | MSE↑ | ±.0002
.0486 | ±.0003
.0374 | ±.0005
.0416
(.0042↑) | ±.0007
.0416
(.0042↑) | ±.0001
**.0469**
(**.0095↑**) | ±.0002
.0322 | ±.0003
.0374
(.0052↑) | ±.0003
.0382
(.0061↑) | ±.0001
**.0452**
(**.0131↑**) | ±.0003
.0087 | ±.0008
.0157
(.0070↑) | ±.0003
.0174
(.0087↑) | ±.0004
**.0409**
(**.0322↑**) |
| | SSIM↓ | ±.0002
.1688 | ±.0135
.2904 | ±.0140
.2430
(.0475↓) | ±.0102
.2236
(.0668↓) | ±.0099
**.1846**
(**.1059↓**) | ±.0042
.3348 | ±.0076
.2654
(.0694↓) | ±.0076
.2454
(.0894↓) | ±.0017
**.1855**
(**.1493↓**) | ±.0114
.8862 | ±.0133
.6782
(.2081↓) | ±.0081
.6369
(.2493↓) | ±.0176
**.2740**
(**.6123↓**) |
| | TCA↓ | ±.0058
.2969 | ±.0276
.3769 | ±.0219
.3157
(.0611↓) | ±.0184
.3125
(.0644↓) | ±.0138
**.2902**
(**.0867↓**) | ±.0049
.3952 | ±.0060
.3375
(.0577↓) | ±.0141
.3334
(.0618↓) | ±.0061
**.2978**
(**.0974↓**) | ±.0076
.6651 | ±.0122
.5770
(.0881↓) | ±.0096
.5780
(.0872↓) | ±.0194
**.3290**
(**.3361↓**) |
| | DeepPSiM↓ | ±.0001
.8344 | ±.0016
.8500 | ±.0018
.8441
(.0059↓) | ±.0017
.8441
(.0059↓) | ±.0010
**.8364**
(**.0136↓**) | ±.0007
.8603 | ±.0009
.8505
(.0098↓) | ±.0011
.8485
(.0118↓) | ±.0002
**.8386**
(**.0217↓**) | ±.0023
.9378 | ±.0023
.9112
(.0265↓) | ±.0013
.9073
(.0305↓) | ±.0024
**.8487**
(**.0890↓**) |
| **ImageNet-100** | MSE↑ | ±.0002
.0438 | ±.0008
.0324 | ±.0013
.0377
(.0053↑) | ±.0011
.0364
(.0040↑) | ±.0003
**.0418**
(**.0094↑**) | ±.0002
.0366 | ±.0002
.0401
(.0034↑) | ±.0002
.0377
(.0011↑) | ±.0002
**.0425**
(**.0059↑**) | ±.0009
.0166 | ±.0007
.0261
(.0095↑) | ±.0015
.0309
(.0143↑) | ±.0005
**.0437**
(**.0272↑**) |
| | SSIM↓ | ±.0016
.3393 | ±.0096
.3877 | ±.0060
.3471
(.0405↓) | ±.0067
.3463
(.0414↓) | ±.0019
**.3385**
(**.0492↓**) | ±.0014
.3469 | ±.0013
.3410
(.0058↓) | ±.0018
.3410
(.0059↓) | ±.0037
**.3335**
(**.0134↓**) | ±.0247
.6541 | ±.0073
.4256
(.2285↓) | ±.0149
.3560
(.2981↓) | ±.0031
**.3371**
(**.3170↓**) |
| | TCA↓ | ±.0022
.0373 | ±.0161
.1242 | ±.0107
.0893
(.0349↓) | ±.0080
.0711
(.0531↓) | ±.0039
**.0445**
(**.0797↓**) | ±.0025
.0615 | ±.0055
.0514
(.0101↓) | ±.0021
.0469
(.0145↓) | ±.0026
**.0388**
(**.0227↓**) | ±.0117
.2689 | ±.0057
.2067
(.0622↓) | ±.0234
.0876
(.1813↓) | ±.0043
**.0416**
(**.2273↓**) |
| | DeepPSiM↓ | ±.0004
.5451 | ±.0073
.5826 | ±.0098
.5755
(.0072↓) | ±.0056
.5556
(.0270↓) | ±.0009
**.5484**
(**.0342↓**) | ±.0017
.5593 | ±.0029
.5541
(.0052↓) | ±.0011
.5505
(.0088↓) | ±.0007
**.5464**
(**.0129↓**) | ±.0107
.6826 | ±.0033
.6440
(.0386↓) | ±.0086
.5838
(.0988↓) | ±.0027
**.5507**
(**.1319↓**) |

(OND); and (5) ExpMI attacks against GNIME. When measuring the defensive capability of RND, OND, and GNIME, each MI network was separately trained using the explanations perturbed by the corresponding defense method. This resembles the real-world adversary in our black-box setting, who has no way of obtaining clean model explanations.

Table 1 compares the performance of GNIME against these experimental settings assuming a target ML service providing Grad, Grad-CAM, and LRP, respectively. Recall from §3 that we assume a strong adversary in possession of auxiliary data of which the distribution is the same as the training dataset of the target model. We thus use the train split of our attack dataset for training GNIME and the test split for measuring the metrics (§5.1). Considering that ExpMI shows better reconstruction quality compared to PredMI in all the cases, exploiting model explanations to perform MI attacks indeed contributes to additional information leakage. Surprisingly, the SSIM score significantly boosted from 0.1688 to 0.8862 when targeting CIFAR-10-LRP. These results demonstrate the necessity of defense against ExpMI attacks. We now compare the performance of GNIME against other two defense settings (RND and OND) that add noise to achieve the same defensive goal. RND corresponds to a naive approach that randomly adds Gaussian noise, which serves as our defense baseline. For OND, instead of training $f_{NG}$, we use the projected gradient descent [18] to optimize the input image towards increasing the loss function of $f_{INV}$. Note that adding huge noise can prevent information leakage, but it would undermine the ability to explain the model's decisions. Hence, for fair comparison, we clip the perturbation magnitude of RND and OND to align with the average perturbation size of GNIME.

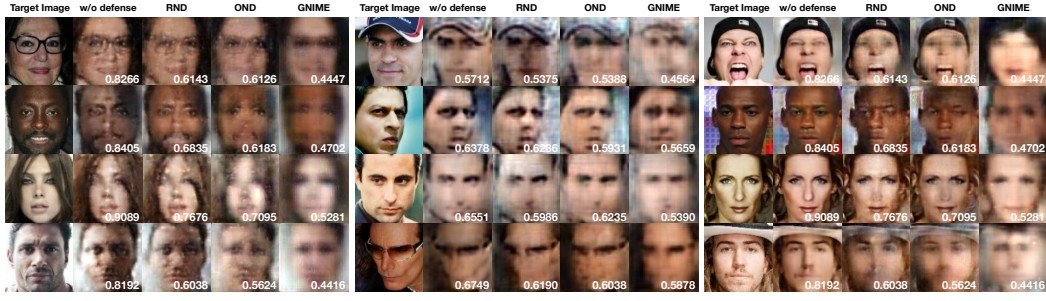

| (a) ExpMI against Grad | (b) ExpMI against Grad-CAM | (c) ExpMI against LRP |

Figure 4: For each explanation type, we compare the ExpMI inversion results of four cases: no defense, RND, OND, and GNIME (columns 2–5). The SSIM score of each reconstructed image compared to the target image (column 1) is marked in white.

As shown in the table, GNIME succeeds in decreasing the information leakage threat for all the explanations and data types. Although RND also slightly alleviates the MI threat by degrading the inversion quality, GNIME significantly outperforms RND in all the cases. Furthermore, GNIME shows superior performance compared to OND as well in all cases. For example, when targeting CelebA-LRP, RND and OND lower the TCA by 0.0588 and 0.0940, respectively. On the other hand, GNIME decreases the TCA by 0.1821, which corresponds to 3.10× of RND and 1.94× of OND. From these observations, we conclude that noise optimized against the $f_{INV}$ fails to generalize on $f_A$, which is independently trained for ExpMI attacks, and it requires a dedicated model $f_{NG}$ to generate effective inversion-resistant noise.

PredMI serves as an upper bound for all the presented defense frameworks, which shows the case when there exists no additional information leakage threat that stems from providing model explanations (§5.1). Considering this upper bound shown in the table, we believe that the degree of GNIME achieving its defensive goal is close to its upper bound. For instance, the DeePSiM score against PredMI and ExpMI targeting CelebA-Grad with GNIME achieves 0.1851 and 0.1880, respectively.

Figure 4 presents the qualitative experimental results. We observe that GNIME outperforms both RND and OND in decreasing the reconstruction quality. When comparing all columns between the third and fifth for each explanation type, GNIME clearly contributes to degrading the quality of reconstructed images; ExpMI still reconstructs most features in the inversion images when the explanations are under the protection of RND or OND, contrary to the inversion images with GNIME.

We further compare the inversion results with a more practical adversary. Here, the adversary cannot train its inversion network $f_A$ with the same dataset as the target model (CelebA). Instead, it draws the training data from another generic distribution. Specifically, we use a subset of FaceScrub to train $f_A$. This auxiliary training set is arguably easier to obtain than the original dataset [29].

Table 2 summarizes the ExpMI attack performance with and without GNIME. The metrics consistently report lower reconstruction performance even when the adversary leverages FaceScrub as the auxiliary dataset. Interestingly, the defensive capability of GNIME was even greater in the practical setting (e.g., using FaceScrub to attack CelebA-LRP, the degradation of TCA due to GNIME increased from 84.82% degradation (0.2147→0.0326) to 99.43% (0.2086→0.0012)). These results demonstrate that GNIME is more effective against the practical adversary of conducting ExpMI attacks.

### 5.3.2 Preserving XAI Functionality

In this section, we further evaluate whether an attribution map $\hat{e}$ noised by GNIME still provides interpretable explanations on the model's decisions, thus preserving its functionality. The XAI methods (e.g., Grad, Grad-CAM, and LRP) produce an attribution map that reflects the degree of importance for each pixel contributing to a target model emitting a decision. Therefore, an input image multiplied by this model explanation ($x \odot e$) erodes unimportant pixels and leaves important pixels intact. That is, we can indirectly evaluate whether the original functionality of a model explanation $e$ is preserved in its perturbed version $\hat{e}$ by comparing the pixels left in $x \odot e$ and $x \odot \hat{e}$ using DeePSiM. We compare the intermediate representation values in the penultimate layer of the target classifier

Table 2: MI attack performance of an adversary exploiting auxiliary dataset (FaceScrub) of which distribution is different from that of the original training set (CelebA).

| Metric | PredMI | Grad | | Grad-CAM | | LRP | |
|---|---|---|---|---|---|---|---|
| | | ExpMI | GNIME | ExpMI | GNIME | ExpMI | GNIME |
| MSE↑ | ±.0011 .0319 | ±.0010 .0169 | ±.0005 .0255 (**.0086**↑) | ±.0017 .0212 | ±.0007 .0252 (**.0040**↑) | ±.0018 .0088 | ±.0007 .0266 (**.0165**↑) |
| SSIM↓ | ±.0217 .4257 | ±.0248 .6218 | ±.0099 .4424 (**.1795**↓) | ±.0330 .5583 | ±.0174 .4317 (**.1266**↓) | ±.0351 .8038 | ±.0056 .4327 (**.3475**↓) |
| TCA↓ | ±.0046 .0171 | ±.0164 .0968 | ±.0004 .0008 (**.0960**↓) | ±.0137 .1014 | ±.0004 .0009 (**.1005**↓) | ±.0168 .2086 | ±.0005 .0012 (**.1970**↓) |
| DeePSiM↓ | ±.0034 .1795 | ±.0105 .2320 | ±.0017 .1795 (**.0525**↓) | ±.0190 .2621 | ±.0030 .1954 (**.0667**↓) | ±.0456 .4560 | ±.0021 .1791 (**.2441**↓) |

Table 3: DeePSiM score between $x \odot e$ and $x \odot \hat{e}$. Values within parentheses indicate the average perturbation magnitude (in MSE).

| Dataset | Grad | Grad-CAM | LRP |
|---|---|---|---|
| CelebA | (.0026) .8107 | (.0046) .7806 | (.0033) .6734 |
| MNIST | (.0039) .9571 | (.0098) .9516 | (.0077) .9298 |
| CIFAR-10 | (.0120) .9450 | (.0176) .9263 | (.0183) .8761 |
| ImageNet-100 | (.0011) .9097 | (.0081) .9263 | (.0010) .9290 |

$f$ for each input: $x \odot e$ and $x \odot \hat{e}$. A higher DeePSiM score indicates a higher resemblance in the high-level features captured by the target model.

Table 3 shows the average DeePSiM value of all $x \odot e$ and $x \odot \hat{e}$ pairs in which each $x$ is from the test split of the attack dataset. The model perceptual similarity between $x \odot e$ and $x \odot \hat{e}$ is at least 0.6734, 0.9298, 0.8761, and 0.9097 for the CelebA, MNIST, CIFAR-10, and ImageNet-100 models, respectively. This indicates that the original functionality of $e$ largely remains intact.

**Perturbation Magnitude.** We analyze the distribution of MSEs between $e$ and $\hat{e}$ pairs for CelebA depicted in Figure 5. The figure shows two triples $(x, e, \hat{e})$, each of which respectively represents a common case and an extreme case of perturbation magnitude (i.e., MSE = 0.0015 and MSE = 0.0055). Note that even the perturbation of the extreme case is hardly perceptible. These results suggest that GNIME applies infinitesimal perturbations to $e$ such that it preserves its original functionality. We include further visualization of perturbed model explanations in the supplementary material.

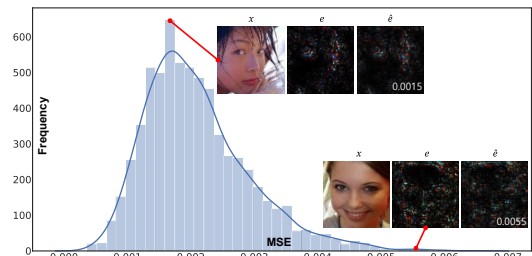

Figure 5: Distribution of GNIME's perturbation magnitude on CelebA-Grad explanations. Values marked white inside $\hat{e}$ indicate the perturbation magnitude (in MSE).

In addition, recall from §4 that one can further vary $\gamma$ in Equation 3 to control the perturbation magnitude. We observed a linear trade-off relationship between the defensive capability and the explanation functionality of GNIME. GNIME service providers can use $\gamma$ of their choice with this in mind. When choosing to use 0.0040 for $\gamma$, GNIME achieves the 0.6340 DeePSiM. However, when limiting the perturbation size to 0.0005, the DeePSiM score increases to 0.8497. Please refer to the supplementary material for further experimental results.

### 5.3.3 Investigating Additional Information Leakage.

Although GNIME significantly lowers the risk of information leakage to the degree to that of PredMI, information leakage still remains to some extent even after its deployment. However, we stress that GNIME focuses on preventing *additional* information leakage that stems from model explanations; therefore, the inversion results of PredMI form an upper bound of our defense. Note that GNIME is capable of obscuring several key attributes in the original images, providing inversion results similar to those of PredMI. For instance, Figure 1 shows that GNIME removes critical features in the reconstructed images, such as sunglasses, cap, or gender information.

We additionally evaluate whether GNIME is able to obscure several key attributes of reconstructed images. Similar to Chen *et al.* [5], we trained attribute classifiers using CelebA and evaluated whether the attribute classifiers is able to correctly predict attributes when the reconstructed images are provided. Specifically, we trained eight attribute classifiers, employing the ResNet50V2 architecture from He *et al.* [12] with its final layer replaced for binary classification. Note that the training set used to train these classifiers is disjoint from that we used in the input reconstruction. We report the F1 score to measure the performance of these attribute classifiers. Figure 6 summarizes the evaluation results. GNIME performed best in terms of deteriorating the attribute prediction performance for all

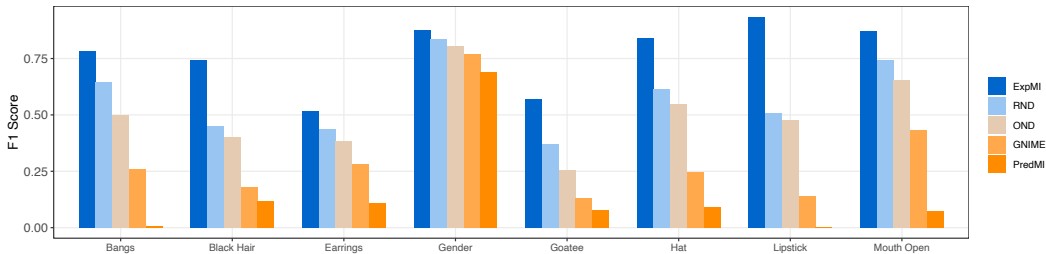

Figure 6: F1 scores measured on eight attribute classifiers using reconstructed images.

attribute types. On average, ExpMI without defense reports an F1 score of 0.7672. Note that GNIME decreases it to 0.3048, which is close to the F1 score of PredMI.

## 6 Discussions and Limitations

**Constrained Obfuscation.** The concept of obfuscating sensitive information while retaining useful information has been explored in various fields in establishing privacy [4, 16] and fairness[30]. Their common approach is to obfuscate given input instances and then jointly train a target model to retain a high performance even for those obfuscated instances. By contrast, GNIME does not directly alter the target model parameters and focuses only on obfuscating model explanations to minimize the MI threat. This enables the deployment of legacy models without any retraining steps while preserving the original model performance.

**Model Extraction.** Zhao *et al.* [32] proposed an ExpMI attack even when a target DNN model does not provide any explanation (ExpMI-3), which we do not assume in the threat model. The core idea of ExpMI-3 is to build a high-performing surrogate model through a successful model extraction (ME) attack and then construct surrogate explanations based on this surrogate model. However, we stress that once the adversary successfully constructs the surrogate model, the surrogate model exposes the target model to numerous white-box attacks, not only limited to the ExpMI threat. We argue that a defense against ExpMI-3 should focus on mitigating the ME threat. We implemented a simple ME defense, reverse sigmoid perturbation (RSP) [15], and evaluated its efficacy in terms of degrading the inversion performance of ExpMI-3; the SSIM score of ExpMI-3 attack against RSP-equipped CelebA was 0.4928, which is comparable to the SSIM of ExpMI attack against GNIME-equipped CelebA-LRP, 0.4960. Further experimental results are given in the supplementary material.

**Black-box Adversary.** Previous MI attacks [5, 31] have been proposed assuming a white-box adversary who can access the parameters and gradients of a target model. By contrast, we assume the black-box adversary conducting ExpMI attacks, following Zhao *et al.* [32]. Therefore, GNIME is designed to mitigate the ExpMI threat in a black-box manner in which a target model returns its explanation and output upon a given input. Considering that many Machine-Learning-as-a-Service (MLaaS) services operate in a black-box setting, we emphasize that addressing the ExpMI threat in this setting is still important. To our knowledge, GNIME is the first defense approach to effectively mitigate the ExpMI threat.

## 7 Conclusion

In this paper, we propose GNIME, the first defense framework against MI attacks using model explanations. During the training of GNIME, we put together the noise generator and inversion networks so that they compete with each other; the noise generator network learns to inject imperceptible noise into model explanations that undermine model inversion, and the inversion network learns to excel at reconstructing the original inputs from model explanations. GNIME then leverages the trained noise generator network for every model explanation upon its release. Our evaluations across diverse datasets and explanation types demonstrate that GNIME is effective in mitigating the privacy threat that MI attacks impose while preserving the original functionality of model explanations.

**Acknowledgements.** This work was supported by the Institute of Information & communications Technology Planning & evaluation (IITP) grant funded by the Korea government (MSIT) (No.2020-0-00153, Penetration Security Testing of ML Model Vulnerabilities and Defense).

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
