# OpenReview forum: "Learning to Generate Inversion-Resistant Model Explanations"
_NeurIPS.cc/2022/Conference — NeurIPS 2022 Accept_

### Official Review · Reviewer_FbTU · 2022-07-11

**Rating:** 6
**Confidence:** 3
**Soundness:** 3 good
**Presentation:** 2 fair
**Contribution:** 3 good

**Summary:**

The authors proposed a defense methodology against model inversion attack, which uses model explanation results to recover the training set. The purposes of this defense method are to minimize the attack success rate and keep the model's interpretability. The authors train noise generator and inversion network to achieve these purposes by using a minimax setting. They evaluated the efficacy of their methodology by measuring defense success rate and interpretability using various datasets.


**Questions:**

1. Is there a threat when the adversary knows the fact that the GNIME is used?

**Limitations:**

Limitation is enoughly discussed in limitation section of main paper.

<minor suggestion>

1. In line 286, please specify the table.
2. Please add the attribution maps for qualitative comparison. (Fig 5. is too small to see the difference.)


---
Thank you for the additional experiments and explanations.

The additional experiments show that GNIME is also working for the higher resolution images from Imagenet(256x256).
However, I expected 512 or 1024 resolution of human figures (e.g., FFHQ) or X-ray.
These datasets are more important for privacy.

The explanation for other adversary scenario is great.
But I am going to maintain the score since the additional experiments are not resolving weakness.

**Strengths And Weaknesses:**

Strength
1. Proposed first defense method against explanation aware model inversion attack.
2. Did the same experiment several times to show the consistency of their method.

Weakness
1. The image data used were limited to low resolution. Since the methodology preserves the privacy of the training set, higher resolution images of human figures or X-rays would be a good choice to check the efficacy of methodology.

---

> ### Author Response · Authors · 2022-08-02
> **Author Response to Reviewer FbTU**
>
> We thank the reviewer for the constructive feedback. We address each concern in the following.
>
>
> **The image data used were limited to low resolution**
>
> We conducted an additional experiment on a dataset that has a higher resolution, extending Table 1. In this experiment, we used ImageNet-100 (consisting of 100 randomly selected classes from ImageNet), which has a size of 256 $\times$ 256. We referenced our model architecture from the model that Zhao et al. [27] used. It consists of four convolutional layers followed by two fully-connected layers. We employed a dropout layer to avoid overfitting.
>
> As shown in the table ([table](https://drive.google.com/uc?export=view&id=1Lr-g8BPNumoAhSPSS8-GLownEO2A_GOT)), GNIME still outperforms RND and OND, degrading the image reconstruction quality to the degree of PredMI. When targeting the ImageNet100 model-LRP, RND and OND lower the TCA by 0.0466 and 0.0573, respectively. By contrast, GNIME decreases the TCA by 0.2036, which corresponds to 4.36$\times$ of RND and 3.54$\times$ of OND. Overall, the performance metrics of GNIME are close to its upper bound, those of PredMI. These results demonstrate that GNIME is still effective in alleviating the model inversion threat on a large-scale dataset, ImageNet-100.
>
> For the qualitative comparison of reconstructed images, please refer to the figure ([figure](https://drive.google.com/uc?export=view&id=1hk_cije-ylxOZtn1TAWeWjeFEujJrAJR)), which depicts the quality of reconstructed images using LRP explanations. The inversion results for other explanation types are given in the updated supplementary material. We will include these experimental results in the revised paper.
>
> &nbsp;
>
> **Is there a threat when the adversary knows the fact that the GNIME is used?**
>
> We appreciate this insightful comment and have contemplated the possible strategies that this adversary can take. This adversary can adopt three different strategies: (1) avoiding the use of perturbed explanations (i.e., utilize only the prediction vectors) for training the inversion model; (2) training a surrogate model in mimicry of the target model to generate clean explanations; or (3) training the inversion model to reconstruct inputs from perturbed explanations.
>
> (1) - This case coincides with PredMI. Considering that the PredMI inversion performance  is even lower than that of GNIME, the adversary has little incentive to take this strategy. Also, GNIME does not focus on addressing the inversion risk due to PredMI.
>
> (2) - The adversary may attempt to create a surrogate model by conducting model extraction attacks. The adversary can directly generate clean model explanations from this surrogate model, as shown in Zhao et al. [27]. Otherwise, the adversary can also use the surrogate model to train a denoiser that removes perturbations in our model explanations. However, in both cases, model extraction should successfully copy the victim model’s functionality into the surrogate model in the first place. Therefore, we believe that this case should be addressed by mitigating the model extraction threat. We have already discussed this case in detail in the limitations section and supplementary material. In Section 4 of the supplementary material, we showed that a simple model extraction defense of leveraging reverse sigmoid perturbation can mitigate this threat to the degree to which GNIME can protect the target model.
>
> (3) - The adversary may try to train an inversion model that can directly reconstruct inputs from perturbed model explanations. Please note that our evaluation in Section 5 already assumes this adversary. We demonstrated that GNIME successfully mitigates the model inversion threat from such an attacker.
>
> We will add the discussions above in the revised paper.
>
> &nbsp;
>
>  **Minor suggestions**
>
> Thank you for the detailed suggestions! We will make all the suggested changes in the revised version.

---

> > ### Author Response · Authors · 2022-08-09
> > **Updated the broken links in the rebuttal.**
> >
> > We realized that our links to the anonymous submission site are unstable, which may make the reviewers unable to access the figures and tables. So, we updated the links above to Google Drive with an anonymous account.
> >
> > For the table that shows the ImageNet-100 performance, please refer to \
> > https://drive.google.com/uc?export=view&id=1Lr-g8BPNumoAhSPSS8-GLownEO2A_GOT
> >
> > For the qualitative comparison of reconstructed images, refer to \
> > https://drive.google.com/uc?export=view&id=1hk_cije-ylxOZtn1TAWeWjeFEujJrAJR

---

### Official Review · Reviewer_aV5M · 2022-07-12

**Rating:** 7
**Confidence:** 4
**Soundness:** 3 good
**Presentation:** 3 good
**Contribution:** 3 good

**Summary:**

This work proposes a defense method that add noise to model explanations to minimize the risk of model inversion attacks, and preserving the interpretabilities of the explanations. They formulate the problem of defense training as a two-player minimax game between the MI attacker and the noise generator.  The MI attacker takes in the explanation and prediction vector, generates the original input, and the noise generator takes in the model explanation and prediction vector and generates a perturbed version of the explanation in order to prevent the MI attacker from leaking information. The proposed framework is able to defense against MI attack according to the results, achieving a larger reconstruction error than other baselines, and it is able to preserve the effectiveness of the model explanations to some extent.

**Questions:**

From the results of the paper, the proposed method achieves better performance than the mentioned baselines. However, the reconstructed image can still reveal some features of the original image. Is there a potential risk that even the revealed partial information may be abused by attackers to reveal privacy of the original image?

The results provide detailed numbers of performance metrics, but it is not clear whether the improvements are significant or not. Is there a better way to show that the improvements are statistically significant?

**Limitations:**

To some extent, but the authors did not mention what other information could be leaked even given the blurry reconstructed images. Ideally, the reconstructed image should be very blurred and cannot be recognized as human face.

**Strengths And Weaknesses:**

originality: the proposal to formulate the defense problem as a min-max game is very interesting and the results show that it works to some extent. The proposed evaluation metrics clearly show the advantage of the proposed method over the baselines such as gaussian noise injection.

quality and clarity: the work's method is clear and experiments include detailed comparisons with several baselines. However, with so many detailed quantitative results, it is not clear whether the achieved results are significant or not. Maybe there could be a better way to present the data so that it is not so messy and could be clearer for readers to understand the results. Plots could be a better option than tables to show the results since they are more straightforward and intuitive. Many experimental details are included in the text, which makes it hard to understand the big picture. Maybe a better way is to have a table summarizing the details of various hyper parameters, and have more subsections each covering one topic.

Significance: the method itself is interesting and valuable to the community as a way to defend against MI attack. However, it is less clear to me how significant are the results. On one hand, the used dataset is relatively small, what about the results on large scale ImageNet dataset? On the other hand, the reconstructed image even after the noise perturbation on the explanation can still reveal some characteristics of the original image. For example, the gender and some facial features on the human face images can be identified, even if the proposed method achieves better metrics than baselines.

---

> ### Author Response · Authors · 2022-08-02
> **Author Response to Reviewer aV5M**
>
> We appreciate your helpful feedback. We answer each question in the following.
>
> **Used dataset is relatively small.**
>
> We conducted an additional experiment on a dataset that has a higher resolution, extending Table 1. In this experiment, we used ImageNet-100 (consisting of 100 randomly selected classes from ImageNet), which has a size of 256 $\times$ 256. We referenced our model architecture from the model that Zhao et al. [27] used. It consists of four convolutional layers followed by two fully-connected layers. We employed a dropout layer to avoid overfitting.
>
> As shown in the table ([table](https://drive.google.com/uc?export=view&id=1Lr-g8BPNumoAhSPSS8-GLownEO2A_GOT)), GNIME still outperforms RND and OND, degrading the image reconstruction quality to the degree of PredMI. When targeting the ImageNet100 model-LRP, RND and OND lower the TCA by 0.0466 and 0.0573, respectively. By contrast, GNIME decreases the TCA by 0.2036, which corresponds to 4.36$\times$ of RND and 3.54$\times$ of OND. Overall, the performance metrics of GNIME are close to its upper bound, those of PredMI. These results demonstrate that GNIME is still effective in alleviating the model inversion threat on a large-scale dataset, ImageNet-100.
>
> For the qualitative comparison of reconstructed images, please refer to the figure ([figure](https://drive.google.com/uc?export=view&id=1hk_cije-ylxOZtn1TAWeWjeFEujJrAJR)), which depicts the quality of reconstructed images using LRP explanations. The inversion results for other explanation types are given in the updated supplementary material. We will include these experimental results in the revised paper.
>
> &nbsp;
>
> **Is there a better way to show that the improvements are statistically significant?**
>
> We now report the inversion results of CelebA in this bar chart ([figure](https://drive.google.com/uc?export=view&id=13bIiyk87B7ZkIq5Ag6cyWMPfewg4KowQ)). For now, we only report TCA, which shows the most prominent differences across different defenses.
>
> To check whether the measured performance differences between GNIME and another defense are statistically significant, we performed two-tailed Mann Whitney U tests and reported p-values for each pair of GNIME and another defense. As the table ([table](https://drive.google.com/uc?export=view&id=1EISU7_qILGTS2EvqPE0jdPFzd9_96yO0)) shows, most U test results are statistically signiﬁcant with p-values less than 0.05, showing that the superior performance of GNIME over other defenses is statistically significant.  Please refer to the supplementary material for more details.
>
> &nbsp;
>
> **What other information could be leaked even given the blurry reconstructed images?**
>
> We acknowledge that GNIME cannot prevent all private information leakage. However, recall from Section 5.3.1 that GNIME focuses on preventing **additional** information leakage that stems from model explanations, and the inversion results of PredMI therefore form an upper bound of our defense. We stress that GNIME is capable of obscuring several key attributes in the original images. For instance, note from Figure 1 that GNIME removes critical features in the reconstructed images, such as sunglasses, a cap, or gender information. From our observations in Figure 1, we believe that the inversion performance of PredMI and ExpMI with GNIME is comparable.
>
> We additionally evaluate whether GNIME can obscure key attributes of reconstructed images using attribute classifiers. Similar to Chen et al. [D], we trained attribute classifiers using CelebA and evaluated whether the attribute classifiers can correctly predict attributes when the reconstructed images are provided. Specifically, we trained eight attribute classifiers, employing the ResNet50V2 architecture from He et al. [E], with its final layer replaced for binary classification. Note that the training set used to train these classifiers is disjoint from that we used in the input reconstruction. We report the F1 score to measure the performance of these attribute classifiers. The figure ([figure](https://drive.google.com/uc?export=view&id=1mgxv1W6LV2jI4EHhypD_0ND-plIv8v1h)) summarizes the evaluation results. GNIME performed best in terms of deteriorating attribute predictions for all attribute types. On average, ExpMI without defense reports an F1 score of 0.7672. Note that GNIME decreases it to 0.3048, which is close to the F1 score of PredMI. Please refer to the supplementary material for further details.
>
> &nbsp;
>
> [D] Si Chen, Mostafa Kahla, Ruoxi Jia, Guo-Jun Qi. Knowledge-Enriched Distributional Model Inversion Attacks. Proceedings of the IEEE/CVF International Conference on Computer Vision (ICCV), pages 16178-16187, 2021.
>
> [E] Kaiming He, Xiangyu Zhang, Shaoqing Ren, Jian Sun. Identity Mappings in Deep Residual Networks. Proceedings of the European Conference on Computer Vision (ECCV), pages 630-645, 2016.

---

> > ### Author Response · Authors · 2022-08-09
> > **Updated the broken links in the rebuttal.**
> >
> > We realized that our links to the anonymous submission site are unstable, which may make the reviewers unable to access the figures and tables. So, we updated the links above to Google Drive with an anonymous account.
> >
> > For the table that shows the ImageNet-100 performance, please refer to \
> > https://drive.google.com/uc?export=view&id=1Lr-g8BPNumoAhSPSS8-GLownEO2A_GOT
> >
> > For the qualitative comparison of reconstructed images, refer to \
> > https://drive.google.com/uc?export=view&id=1hk_cije-ylxOZtn1TAWeWjeFEujJrAJR
> >
> > For the comparison of CelebA inversion performance in the form of a bar chart, refer to \
> > https://drive.google.com/uc?export=view&id=13bIiyk87B7ZkIq5Ag6cyWMPfewg4KowQ
> >
> > For the p-value table shows the statistical significance, refer to \
> > https://drive.google.com/uc?export=view&id=1EISU7_qILGTS2EvqPE0jdPFzd9_96yO0
> >
> > For the new table that shows the GNIME’s performance of mitigating information leakage, please refer to \
> > https://drive.google.com/uc?export=view&id=1mgxv1W6LV2jI4EHhypD_0ND-plIv8v1h

---

> > ### Comment · Reviewer_aV5M · 2022-08-09
> > **Follow up**
> >
> > I really appreciate the additional work and explanations by the authors. I keep my rating as the paper is outstanding and my questions are well explained.

---

### Official Review · Reviewer_GkYa · 2022-07-14

**Rating:** 8
**Confidence:** 5
**Soundness:** 4 excellent
**Presentation:** 4 excellent
**Contribution:** 4 excellent

**Summary:**

This paper proposes a perturbation-based approach to prevent an adversary from leveraging model explanations to perform model inversion attacks. In particular, the authors apply this defence mechanism to three explanation types: Grad, Grad-CAM, and LRP. The proposed adversarial training framework consists of a model inversion network that performs the inversion attack and a generator network that learn to create perturbations for explanations. Experiments are performed to demonstrate the proposed approach's ability to reduce the efficacy of explanation-aware model inversion attacks.

**Questions:**

- Please add more discussion on the risk that noisy explanations can lead to manipulated explanations.
- If possible, report the additional score like the SSIM of $x \odot e$ and  $x \odot \hat{e}$ to provide more arguments for the preservation of XAI functionality.

**Limitations:**

Nothing to report.

**Strengths And Weaknesses:**

**Strengths**

-   Several evaluation metrics to compare the original input $x$ with the reconstructed input $\hat{x}$, namely MSE, SSIM, DeePSIM, and TCA
- The proposed defence mechanism is evaluated against state-of-the-art prediction-based and explanation-aware model inversion attacks
- Experiments are performed on several real-world datasets
- Realistic assumptions on the adversary knowledge have been investigated

**Weaknesses**

My main concern is the lack of discussion on the manipulability of saliency maps. Several research works have shown (e.g., [a]) that these types of explanation techniques can be arbitrarily manipulated. While the paper is not concerned with adversarial manipulation of explanations, the noise added to the original explanations can also unintentionally leads to manipulated explanations that fairwashed the original decisions of the target model.

[a] - Explanations can be manipulated and geometry is to blame. (Dombrowski et al., 2019)

---

> ### Author Response · Authors · 2022-08-02
> **Author Response to Reviewer GkYa**
>
> We thank the reviewer for the constructive feedback. We address each concern raised by the reviewer below.
>
> **Risk that noisy explanations can lead to manipulated explanations.**
>
> We understand that the reviewer is concerned with the case in which the noise generator overwrites the critical features of model explanations; thus, perturbed model explanations lose their interpretability. We acknowledge that, in the worst case, our model explanations $\hat{e}$ might look similar to the manipulated explanations in Dombrowski et al. [C]. We will definitely discuss the reviewer's concern about the manipulated explanations in the
> paper. To alleviate this concern, we empirically evaluated whether the perturbed model explanations still preserve their functionality in Section 5.3.2. We measured the DeepSiM between $x \odot e$ and $x \odot \hat{e}$. This metric illustrates the similarity of high-level features captured by the neural network between the key pixels selected by clean and perturbed model explanations. Table 3 demonstrates that there exists a high similarity. The table demonstrates that the model perceptual similarity between $x \odot e$ and $x \odot \hat{e}$ is at least 0.6734, 0.9298, and 0.8761 for the CelebA, MNIST, and CIFAR-10 models, respectively.
>
> In addition, we evaluated the distribution of added perturbation sizes. Figure 5 shows that the sizes of added perturbations are negligible, and the added perturbations do not impair the XAI functionality. As requested by the reviewer, we further measured the SSIM between $x \odot e$ and $x \odot \hat{e}$ in the table below. DeePSiM and SSIM represent the perceptual similarity in the perspective of the target neural network and humans, respectively.
>
> &nbsp;
>
> | **Dataset** | **Metric** | **Grad** | **Grad-CAM** | **LRP** |
> |------------------|-------------|----------|--------------|---------|
> | **CelebA** | **DeePSiM** | 0.8107 | 0.7806 | 0.6734 |
> | | **SSIM** | 0.7862 | 0.7253 | 0.7948 |
> | **MNIST** | **DeePSiM** | 0.9571 | 0.9516 | 0.9298 |
> | | **SSIM** | 0.8975 | 0.9499 | 0.9014 |
> | **CIFAR-10** | **DeePSiM** | 0.9450 | 0.9263 | 0.8761 |
> | | **SSIM** | 0.7366 | 0.7689 | 0.6611 |
> | **ImageNet-100** | **DeePSiM** | 0.9097 | 0.9263 | 0.9290 |
> | | **SSIM** | 0.8687 | 0.7711 | 0.8434 |
>
> &nbsp;
>
> **Report the additional score like the SSIM of $x \odot e$ and $x \odot \hat{e}$**
>
> We report the SSIM between $x \odot e$ and $x \odot \hat{e}$ in the table above. Please note from the table that $x \odot e$ and $x \odot \hat{e}$ also have high structural similarity. Based on these observations, we believe that attribution maps $\hat{e}$ provide interpretable explanations on the model’s decisions. We will include this evaluation result in Table 3.
>
> &nbsp;
> &nbsp;
>
> [C] Ann-Kathrin Dombrowski, Maximillian Alber, Christopher Anders, Marcel Ackermann, Klaus-Robert Muller, Pan Kessel. Explanations can be manipulated and geometry is to blame. Advances in Neural Information Processing Systems (NeurIPS), 2019

---

> > ### Comment · Reviewer_GkYa · 2022-08-09
> > **Update of my score**
> >
> > Great!
> > Thanks for the updates.
> > I have increased my score accordingly.

---

### Official Review · Reviewer_KFM9 · 2022-07-18

**Rating:** 6
**Confidence:** 5
**Soundness:** 3 good
**Presentation:** 4 excellent
**Contribution:** 3 good

**Summary:**

The authors consider the problem of model inversion using an auxiliary dataset identically distributed to the training data, its predictions under the model as well as their explanation maps under Grad, Grad-Cam, and LRP saliency map. Given a trained target classifier, their approach consists in setting up an adversarial game between a noise injection and model inversion networks. The noise injection network returns a censored version of the explanation by minimizing the euclidean distance between the explanation and its censored version while maximizing the euclidean distance between the original image and its reconstruction by the inversion model network. The inversion model network aims to predict the original images from the target model predictions, explanations and the injection network censored explanation.

**Questions:**

The paper would be substantially strengthened and find a wider audience by considering tabular and text data.
Considering large scale, complex image data would give an opportunity for the method to set itself apart even more.
A theoretical analysis of the algorithm convergence, even in a non-parametric setting and just at optimality, could shed light on the relationship between the noise injection and model inversion network after training.

**Limitations:**

Yes.

**Strengths And Weaknesses:**

Strengths:

* The paper is clearly written and easy to follow.
* The authors consider a wide range of baselines and quantitative evaluation metric.
* The author quantitatively show that censored explanations remain interpretable.

Weaknesses:
* The paper only consider small scale image data.
* It's not clear whether the algorithm converge or not.

---

> ### Author Response · Authors · 2022-08-02
> **Author Response to Reviewer KFM9**
>
> We thank the reviewer for the constructive feedback. We address each concern in the following.
>
> **Considering large scale, complex image data would give an opportunity for the method to set itself apart even more.**
>
> We conducted an additional experiment on a dataset that has a higher resolution, extending Table 1. In this experiment, we used ImageNet-100 (consisting of 100 randomly selected classes from ImageNet), which has a size of 256 $\times$ 256. We referenced our model architecture from the model that Zhao et al. [27] used. It consists of four convolutional layers followed by two fully-connected layers. We employed a dropout layer to avoid overfitting.
>
> &nbsp;
>
> |  |  |  | **Grad** |  |  |  | **Grad-CAM** |  |  |  | **LRP** |  |  |  |  |
> |---|---|---|---|---|---|---|---|---|---|---|---|---|---|---|---|
> |  | **Metric** | **PredMI** | **ExpMI** | **RND** | **OND** | **GNIME** | **ExpMI** | **RND** | **OND** | **GNIME** | **ExpMI** | **RND** | **OND** | **GNIME** |  |
> |  | **MSE ↑** | .0434 | .0356 | .0377 | .0402 | **.0418** | .0362 | .0385 | .0408 | **.0424** | .0181 | .0241 | .0268 | **.0440** |  |
> |  |  |  |  | (.0021↑) | (.0046↑) | **(.0062↑)** |  | (.0023↑) | (.0046↑) | **(.0062↑)** |  | (.0060↑) | (.0087↑) | **(.0259↑)** |  |
> |  | **SSIM ↓** | .3319 | .3633 | .3527 | .3429 | **.3381** | .3459 | .3432 | .3418 | **.3349** | .6193 | .4291 | .4146 | **.3344** |  |
> |  |  |  |  | (.0106↓) | (.0204↓) | **(.0252↓)** |  | (.0027↓) | (.0041↓) | **(.0110↓)** |  | (.1902↓) | (.2047↓) | **(.2849↓)** |  |
> |  | **TCA ↓** | .0407 | .0960 | .0603 | .0530 | **.0503** | .0597 | .0537 | .0480 | **.0420** | .2483 | .2017 | .1910 | **.0447** |  |
> |  |  |  |  | (.0357↓) | (.0430↓) | **(.0457↓)** |  | (.0060↓) | (.0117↓) | **(.0177↓)** |  | (.0466↓) | (.0573↓) | **(.2036↓)** |  |
> |  | **DeePSiM ↓** | .5451 | .5649 | .5589 | .5608 | **.5491** | .5572 | .5540 | .5518 | **.5466** | .6689 | .6417 | .6315 | **.5510** |  |
> |  |  |  |  | (.0060↓) | (.0041↓) | **(.0158↓)** |  | (.0032↓) | (.0054↓) | **(.0106↓)** |  | (.0272↓) | (.0374↓) | **(.1179↓)** |  |
>
> &nbsp;
>
> As shown in the table above (see [table](https://drive.google.com/uc?export=view&id=1Lr-g8BPNumoAhSPSS8-GLownEO2A_GOT) for LateX view), GNIME still outperforms RND and OND, degrading the image reconstruction quality to the degree of PredMI. When targeting the ImageNet100 model-LRP, RND and OND lower the TCA by 0.0466 and 0.0573, respectively. By contrast, GNIME decreases the TCA by 0.2036, which corresponds to 4.36$\times$ of RND and 3.54$\times$ of OND. Overall, the performance metrics of GNIME are close to its upper bound, those of PredMI. These results demonstrate that GNIME is still effective in alleviating the model inversion threat on a large-scale dataset, ImageNet-100.
>
> For the qualitative comparison of reconstructed images, please refer to the figure ([figure](https://drive.google.com/uc?export=view&id=1hk_cije-ylxOZtn1TAWeWjeFEujJrAJR)), which depicts the quality of reconstructed images using LRP explanations. The inversion results for other explanation types are given in the updated supplementary material. We will include these experimental results in the revised paper.
>
> &nbsp;
>
> **Theoretical analysis of the algorithm convergence**
>
> We will discuss the convergence of our loss function in the revised paper. Unlike the original GAN, it is not straightforward to prove that the INV network obtains its optimal solution for any given noise generator. The multiple objectives using model explanations also make it difficult to perform a theoretical analysis. Thus, similar to previous works [A,B] demonstrating their empirical loss convergence, we conduct an empirical analysis. We show that GNIME becomes better at injecting noise that undermines the inversion performance as the epoch in training the NG network increases. This figure ([figure](https://drive.google.com/uc?export=view&id=1PB0USz8yv3_V0tY5Gpo6U9CCfru2Ndb8)) demonstrates that the NG network in GNIME becomes better at deteriorating the inversion performances (i.e., MSE, SSIM, TCA, and DeepPSiM) as the epoch increases and starts to plateau after 500 epochs.
>
> &nbsp;
> &nbsp;
>
> [A] Divyam Madaan, Jinwoo Shin, Sung Ju Hwang. Learning to Generate Noise for Multi-Attack Robustness. Proceedings of the 38th International Conference on Machine Learning (ICML), pages 7279-7289, 2021.
>
> [B] Yuanhao Cai, Xiaowan Hu, Haoqian Wang, Yulun Zhang, Hanspeter Pfister, Donglai Wei. Learning to Generate Realistic Noisy Images via Pixel-level Noise-aware Adversarial Training. Advances in Neural Information Processing Systems (NeurIPS), pages 3259--3270, 2021.

---

> > ### Author Response · Authors · 2022-08-09
> > **Updated the broken links in the rebuttal.**
> >
> > We realized that our links to the anonymous submission site are unstable, which may make the reviewers unable to access the figures and tables. So, we updated the links above to Google Drive with an anonymous account.
> >
> > For the table that shows the ImageNet-100 performance, please refer to \
> > https://drive.google.com/uc?export=view&id=1Lr-g8BPNumoAhSPSS8-GLownEO2A_GOT
> >
> > For the qualitative comparison of reconstructed images, refer to \
> > https://drive.google.com/uc?export=view&id=1hk_cije-ylxOZtn1TAWeWjeFEujJrAJR
> >
> > For the graph figure that shows the convergence, refer to \
> > https://drive.google.com/uc?export=view&id=1PB0USz8yv3_V0tY5Gpo6U9CCfru2Ndb8

---

### Meta-Review · Area_Chair_Mthy · 2022-08-23

**Recommendation:** Accept
**Confidence:** Certain

**Metareview:**

This paper proposes the GNIME method that defends against explanation-aware model inversion attacks by adversarially manipulating the saliency map. The reviews are mostly positive. The (minor) concerns and suggestions are

- The paper lacks a theoretical analysis of the adversarial framework. A theoretical analysis even at optimality will improve the paper. However, this is not a major concern as the experiment results are quite good.

- The idea of adversarially manipulating a representation (here a saliency map) so as to remove sensitive factors of variation while preserving as much information as possible, is not new. For example, these adversarial frameworks have been studied in the context for learning fair representations, and in many other areas. I suggest including a discussion about this in the final manuscript.


**Award:**

No

---

### Decision · Program_Chairs · 2022-09-14

Accept